# Introducing Operator-Potential Heuristics for Symbolic Search

**Daniel Fišer**[1,2]**, Álvaro Torralba**[3]**, Jörg Hoffmann**[1]

[1] Saarland University, Saarland Informatics Campus, Saarbrücken, Germany
[2] Czech Technical University in Prague, Faculty of Electrical Engineering, Czech Republic
[3] Aalborg University, Denmark
danfis@danfis.cz, alto@cs.aau.dk,hoffmann@cs.uni-saarland.de

## Abstract

Symbolic search, using Binary Decision Diagrams (BDDs) to represent sets of states, is a competitive approach to optimal planning. Yet heuristic search in this context remains challenging. The many advances on admissible planning heuristics are not directly applicable, as they evaluate one state at a time. Indeed, progress using heuristic functions in symbolic search has been limited and even very informed heuristics have been shown to be detrimental. Here we show how this connection can be made stronger for LP-based potential heuristics. Our key observation is that, for this family of heuristic functions, the change of heuristic value induced by each operator can be precomputed. This facilitates their smooth integration into symbolic search. Our experiments show that this can pay off significantly: we establish a new state of the art in optimal symbolic planning.

## 1 Introduction

A[*] search with admissible heuristics and symbolic search are currently the two main contenders for the state of the art in cost-optimal planning. In principle, these are two orthogonal enhancements of a vanilla search algorithm. On the one hand, admissible heuristics aim to reduce the number of explored states. On the other hand, symbolic search uses Binary Decision Diagrams (BDDs) (Bryant 1986) to efficiently represent and manipulate sets of states, greatly speeding up exhaustive search. A natural idea is to combine the two, and indeed that idea has been presented decades ago in the BDDA[*] algorithm (Edelkamp and Reffel 1998; Edelkamp 2002).

Yet that combination has not been an unqualified success. For a heuristic to be effective in symbolic search, two properties are required: (1) it must be possible to efficiently evaluate sets of states represented as BDDs, without evaluating the heuristic on each represented state individually; and (2) it must induce a good partitioning, so that sets of states with the same $g$- and $h$-value can be efficiently represented as BDDs. Property (1) is fulfilled by some of the strongest heuristics for explicit-state search (e.g. symbolic PDBs (Kissmann and Edelkamp 2011; Franco et al. 2017; Torralba, López, and Borrajo 2018)) so they can be used in BDDA[*]. However, it has been shown that even very informative heuristics can be detrimental in symbolic search (Speck, Geißer, and Mattmüller 2020), when they do not fulfill property (2). The main reason is that reducing the amount of expanded states may be detrimental if the underlying BDD representation is less efficient. Due to all this, symbolic bidirectional blind search (without heuristics) is considered the dominant symbolic search approach, and the use of heuristic search in this context has lost traction.

Here we challenge this trend by showing that potential heuristics (Pommerening et al. 2015) yield fresh synergy between heuristic and symbolic search. Such heuristics assign a numeric value (a potential) to each fact of the planning task, in a way so that the sum of the potentials of the facts true in a state is an admissible estimate of the state's goal distance. As we show, potential heuristics are particularly well suited for combination with symbolic search.

Our key observation is that potentials can be computed for each operator rather than for each fact. Such *operator* potentials combine synergically with symbolic search as they have property (1): It turns out that under certain conditions, the operator potential of an operator $o$ is equal to the difference in heuristic values $h(s') - h(s)$ for any state transition $s \rightarrow s'$ induced by the operator $o$. This enables us to efficiently encode potential heuristic information in symbolic search without having to compute the heuristic value of each state during the search (Jensen, Veloso, and Bryant 2008). The main difficulty in doing so is that these operator potentials are real (floating-point) numbers, which can lead to rounding and precision issues. Naively rounding these values may lead to an inconsistent heuristic. We show that this can be dealt with by rounding operator potentials within the integer linear program (ILP) that derives the potential heuristics.

Our empirical analysis shows that potential heuristics also fulfill property (2). That is, they not only reduce the number of explored states, but also lead to improvements on the number of BDD nodes on average. This makes potential heuristic very helpful in symbolic search across a large number of benchmark domains. Overall, symbolic forward search with potential heuristics soundly outperforms symbolic bidirectional search, thus establishing a new state of the art in optimal symbolic planning.

## 2 Preliminaries

We consider the finite domain representation (FDR) of planning tasks (Bäckström and Nebel 1995). An **FDR planning**

**task** $\Pi$ is specified by a tuple $\Pi = \langle \mathcal{V}, \mathcal{O}, I, G \rangle$. $\mathcal{V}$ is a finite set of **variables**, each variable $V \in \mathcal{V}$ has a finite **domain** $\mathrm{dom}(V)$. A **fact** $\langle V, v \rangle$ is a pair of a variable $V \in \mathcal{V}$ and one of its values $v \in \mathrm{dom}(V)$. The set of all facts is denoted by $\mathcal{F} = \{\langle V, v \rangle \mid V \in \mathcal{V}, v \in \mathrm{dom}(V)\}$, and the set of facts of variable $V$ is denoted by $\mathcal{F}_V = \{\langle V, v \rangle \mid v \in \mathrm{dom}(V)\}$. A **partial state** $p$ is a variable assignment over some variables $\mathrm{vars}(p) \subseteq \mathcal{V}$. We write $p[V]$ for the value assigned to the variable $V \in \mathrm{vars}(p)$ in the partial state $p$. We also identify $p$ with the set of facts contained in $p$, i.e., $p = \{\langle V, p[V] \rangle \mid V \in \mathrm{vars}(p)\}$. A partial state $s$ is a **state** if $\mathrm{vars}(s) = \mathcal{V}$. $I$ is an **initial state**. $G$ is a partial state called **goal**, and a state $s$ is a **goal state** iff $G \subseteq s$. Let $p, t$ be partial states. We say that $t$ **extends** $p$ if $p \subseteq t$.

$\mathcal{O}$ is a finite set of **operators**, each operator $o \in \mathcal{O}$ has a precondition $\mathrm{pre}(o)$ and effect $\mathrm{eff}(o)$, which are partial states over $\mathcal{V}$, and a cost $\mathrm{cost}(o) \in \mathbb{R}_0^+$. An operator $o$ is **applicable** in a state $s$ iff $\mathrm{pre}(o) \subseteq s$. The **resulting state** of applying an applicable operator $o$ in a state $s$ is another state $o[\![s]\!]$ such that $o[\![s]\!][V] = \mathrm{eff}(o)[V]$ for every $V \in \mathrm{vars}(\mathrm{eff}(o))$, and $o[\![s]\!][V] = s[V]$ for every $V \in \mathcal{V} \setminus \mathrm{vars}(\mathrm{eff}(o))$. We also assume that for every $V \in \mathrm{vars}(\mathrm{pre}(o)) \cap \mathrm{vars}(\mathrm{eff}(o))$ it holds that $\mathrm{pre}(o)[V] \neq \mathrm{eff}(o)[V]$.

Given a non-negative integer $n \in \mathbb{Z}_0^+$, $[n]$ denotes the set $\{1, \ldots, n\}$ with $[0]$ defined as an empty set. A sequence of operators $\pi = \langle o_1, \ldots, o_n \rangle$ is applicable in a state $s_0$ if there are states $s_1, \ldots, s_n$ such that $o_i$ is applicable in $s_{i-1}$ and $s_i = o_i[\![s_{i-1}]\!]$ for $i \in [n]$. The resulting state of this application is $\pi[\![s_0]\!] = s_n$ and $\mathrm{cost}(\pi) = \sum_{i=1}^{n} \mathrm{cost}(o_i)$ denotes the cost of this sequence of operators. A sequence of operators $\pi$ is called an $s$-**plan** iff $\pi$ is applicable in a state $s$ and $\pi[\![s]\!]$ is a goal state. An $s$-plan $\pi$ is called **optimal** if its cost is minimal among all $s$-plans.

A state $s$ is **reachable** if there exists an operator sequence $\pi$ applicable in $I$ such that $\pi[\![I]\!] = s$. Otherwise, we say that $s$ is unreachable. The set of all reachable states is denoted by $\mathcal{R}$. An operator $o$ is **reachable** iff it is applicable in some reachable state. A state $s$ is a **dead-end state** iff $G \not\subseteq s$ and there is no $s$-plan. A set of facts $M \subseteq \mathcal{F}$ is a **mutex** if $M \not\subseteq s$ for every reachable state $s \in \mathcal{R}$.

A **heuristic** $h : \mathcal{R} \mapsto \mathbb{R} \cup \{\infty\}$ estimates the cost of optimal $s$-plans. The **optimal heuristic** $h^\star(s)$ maps each reachable state $s$ to the cost of the optimal $s$-plan or to $\infty$ if $s$ is a dead-end state. A heuristic $h$ is called (a) **admissible** iff $h(s) \leq h^\star(s)$ for every reachable state $s \in \mathcal{R}$; (b) **goal-aware** iff $h(s) \leq 0$ for every reachable goal state $s$; and (c) **consistent** iff $h(s) \leq h(o[\![s]\!]) + \mathrm{cost}(o)$ for all reachable states $s \in \mathcal{R}$ and operators $o \in \mathcal{O}$ applicable in $s$. It is well-known that goal-aware and consistent heuristics are also admissible. In the context of heuristic search, $h$-value of a state node $s$ refers to the heuristic value of $s$, $g$-value to the cost of the sequence of operators leading to $s$, and $f$-value is simply a sum of $h$-value and $g$-value.

## 3 Symbolic Search Background

Explicit state-space search operates on individual states, whereas symbolic search (McMillan 1993) works on sets of states represented by their characteristic functions. A characteristic function $f_S$ of a set of states $S$ is a Boolean function assigning 1 to states that belong to $S$ and 0 to states that do not belong to $S$. Operations like the union ($\cup$), intersection ($\cap$), and complement of sets of states correspond to the disjunction ($\vee$), conjunction ($\wedge$), and negation ($\neg$) of their characteristic functions, respectively. Binary Decision Diagrams (BDDs) (Bryant 1986) are a efficient data-structure to represent Boolean functions in the form of a directed acyclic graph. The size of a BDD is the number of nodes in this representation. The main advantage of using BDDs is that often a BDD is much smaller than the number of states it represents. In fact, BDDs can be exponentially smaller, as certain sets containing exponentially many states can be represented by BDDs of polynomial size (Edelkamp and Kissmann 2008). Most operations on BDDs take only polynomial time in the size of the BDD, which enables the efficient manipulation of large sets of states.

The most prominent implementation of a symbolic heuristic search in the context of automated planning is BDDA$^*$ (Edelkamp and Reffel 1998) which is a variant of A$^\star$ (Hart, Nilsson, and Raphael 1968) using BDDs to represent sets of states. In BDDA$^*$, operators of planning tasks are represented as transition relations, also using BDDs. A *transition relation* (TR) of an operator $o$ is a characteristic function of pairs of states $(s, o[\![s]\!])$ for all states $s$ such that $s$ is (possibly) reachable and $o$ is applicable in $s$. Having a TR $T_o$ for every operator $o \in \mathcal{O}$, we can construct a TR of a set of operators with the same cost $c$ as $T_c = \bigvee_{o \in \mathcal{O}, \mathrm{cost}(o) = c} T_o$. As the size of $T_c$ may be exponential in the number of operators with cost $c$, in practice, it is often a good idea to use *disjunctive partitioning*. Disjunctive partitioning represents $T_c$ with as few BDDs as possible while keeping the size at bay (Jensen, Veloso, and Bryant 2008; Torralba et al. 2017). Moreover, mutexes can be used for a more accurate approximation of states that are reachable (Torralba et al. 2017).

Like A$^*$, BDDA$^*$ expands states by ascending order of their $f$-value. To take advantage of the symbolic representation, BDDA$^*$ represents all states with the same $g$ and $h$ value in a single BDD $S_{g,h}$ (disjunctive partitioning of $S_{g,h}$ can also be used). Given a set of states $S_{g,h}$ and a TR $T_c$, $\mathrm{image}(S_{g,h}, T_c)$ computes the set of successor states reachable from any state in $S$ by applying any operator represented by $T_c$.[1] The $g$-value of the resulting set of successor states is then simply the $g$-value of $S_{g,h}$ plus $c$. These successor states have to be split according to their $h$ value. This can usually be performed efficiently (e.g. with symbolic PDBs (Kissmann and Edelkamp 2011)) by representing the heuristic as a BDD $S_h$ per heuristic value that represents the set of states with that value and performing a conjunction.

GHSETA$^*$ (Jensen, Veloso, and Bryant 2008) encodes the heuristic function as part of the transition relation, creating multiple TRs depending on the impact of the operators on heuristic value. This is a very efficient way of evaluating the heuristics within symbolic search. However, up to now, all heuristics known to be suitable for this representation were

---

[1]The details how the function $\mathrm{image}$ works are not important here—Torralba et al. (2017) provide a detailed description.

either non-informative, inadmissible, or domain dependent.

## 4 Potential Heuristics Background

Potential heuristics, introduced by Pommerening et al. (2015), assign a numerical value to each fact, and the heuristic value for a state $s$ is then simply a sum of the potentials of all facts in $s$.

**Definition 1.** Let $\Pi$ denote a planning task with facts $\mathcal{F}$. A **potential function** is a function $\mathsf{P} : \mathcal{F} \mapsto \mathbb{R}$. A **potential heuristic** for $\mathsf{P}$ maps each state $s \in \mathcal{R}$ to the sum of potentials of facts in $s$, i.e., $h^{\mathsf{P}}(s) = \sum_{f \in s} \mathsf{P}(f)$.

We will leverage prior work on so-called disambiguation (Alcázar et al. 2013) to strengthen potential heuristics (Fišer, Horčík, and Komenda 2020). A disambiguation of a variable $V$ for a given set of facts $p$ is simply a set of facts $F \subseteq \mathcal{F}_V$ from variable $V$ such that every reachable state extending $p$ contains one of the facts from $F$.

**Definition 2.** Let $\Pi$ denote a planning task with facts $\mathcal{F}$ and variables $\mathcal{V}$, let $V \in \mathcal{V}$ denote a variable, and let $p$ denote a partial state. A set of facts $F \subseteq \mathcal{F}_V$ is called a **disambiguation of $V$ for $p$** if for every reachable state $s \in \mathcal{R}$ such that $p \subseteq s$ it holds that $F \cap s \neq \emptyset$ (i.e., $\langle V, s[V] \rangle \in F$).

Clearly, every $\mathcal{F}_V$ is a disambiguation of $V$ for all possible partial states, and if $\langle V, v \rangle \in p$ and there exists a reachable state extending $p$, then $\{\langle V, v \rangle\}$ is a disambiguation of $V$ for $p$. Moreover, if the disambiguation of $V$ for $p$ is an empty set (for any $V$), then all states extending $p$ are unreachable. Therefore, we can use empty disambiguations to determine unsolvability of planning tasks (if $G$ extends $p$), or to prune unreachable operators (if a precondition of the operator extends $p$). So, from now on we will consider only non-empty disambiguations. Fišer, Horčík, and Komenda (2020) showed how to use mutexes to find disambiguations, so here we will assume we already have disambiguations inferred. Furthermore, to simplify the notation, we introduce a disambiguation map.

**Definition 3.** A mapping $\mathcal{D} : (\mathcal{O} \times \mathcal{V}) \cup \mathcal{V} \mapsto 2^{\mathcal{F}}$ is called a **disambiguation map** if (i) for every operator $o \in \mathcal{O}$ and every variable $V \in \text{vars}(\text{eff}(o))$ it holds that $\mathcal{D}(o, V) \subseteq \mathcal{F}_V$ is a disambiguation of $V$ for $\text{pre}(o)$ such that $|\mathcal{D}(o, V)| \geq 1$; and (ii) for every variable $V \in \mathcal{V}$ it holds that $\mathcal{D}(V) \subseteq \mathcal{F}_V$ is a disambiguation of $V$ for $G$ such that $|\mathcal{D}(V)| \geq 1$.

Now we can state sufficient conditions for the potential heuristic to be admissible, which we will need later on.

**Theorem 4.** *(Fišer, Horčík, and Komenda 2020) Let $\Pi = \langle \mathcal{V}, \mathcal{O}, I, G \rangle$ denote a planning task with facts $\mathcal{F}$, and let $\mathsf{P}$ denote a potential function, and let $\mathcal{D}$ denote a disambiguation map.*
*If*

$$\sum_{V \in \mathcal{V}} \max_{f \in \mathcal{D}(V)} \mathsf{P}(f) \leq 0 \tag{1}$$

*and for every operator $o \in \mathcal{O}$ it holds that*

$$\sum_{V \in \text{vars}(\text{eff}(o))} \max_{f \in \mathcal{D}(o, V)} \mathsf{P}(f) - \sum_{f \in \text{eff}(o)} \mathsf{P}(f) \leq \text{cost}(o), \tag{2}$$

*then the potential heuristic for $\mathsf{P}$ is admissible.*

The conditions from Theorem 4 can be formulated as constraints of a linear program (LP) and any solution (for any objective function) to such LP provides potentials for an admissible potential heuristic. So far, potential heuristics have been used as described in Definition 1, i.e., each fact gets assigned a potential value and the heuristic value for a state $s$ is the sum of potentials of all facts in $s$.

## 5 Operator-Potential Heuristics

Our key observation is that potentials can also be designed in a different way, yielding a new synergy with symbolic search: We can assign a potential to each operator and compute an admissible heuristic value for a state $s$ reached by a sequence of operators $\pi$ as a sum of operator potentials of all operators in $\pi$. We start by introducing an *operator-potential function*.

**Definition 5.** Given a potential function $\mathsf{P}$, and a disambiguation map $\mathcal{D}$, a function $\mathsf{Q} : \mathcal{O} \mapsto \mathbb{R}$ is called an **operator-potential function** for $\mathsf{P}$ and $\mathcal{D}$ if

$$\mathsf{Q}(o) = \sum_{f \in \text{eff}(o)} \mathsf{P}(f) - \sum_{V \in \text{vars}(\text{eff}(o))} \max_{f \in \mathcal{D}(o, V)} \mathsf{P}(f) \tag{3}$$

for every operator $o \in \mathcal{O}$.

Note that the value of $\mathsf{Q}(o)$ is just the value of the left hand side of Eq. (2) with the opposite sign. Or in other words, the operator-potential function for an operator $o$ gives us the lower bound on the change of the heuristic value of the corresponding potential heuristic for the given potential function $\mathsf{P}$ and disambiguation map $\mathcal{D}$. Also note that we can use a different disambiguation map for the inference of potential function $\mathsf{P}$ and for the operator-potential function $\mathsf{Q}$, but we do not see a reason to do that, because we would always want to use a disambiguation as strong as possible.

Next, we show that, if the disambiguation map $\mathcal{D}$ maps every operator $o$ and every effect variable $V \in \text{vars}(\text{eff}(o))$ to a singleton, then $h^{\mathsf{P}}(I) + \sum_{i \in [n]} \mathsf{Q}(o_i) = h^{\mathsf{P}}(s)$ for every sequence of operators $\pi = \langle o_1, \ldots, o_n \rangle$ such that $\pi[\![I]\!] = s$. That is, as long as the preconditions on the variables affected by any operator $o$ are known precisely, the potential heuristic value for any state $s$ can be computed as the potential heuristic value for the initial state plus the sum of operator potentials of operators from any sequence of operators $\pi$ leading to $s$. Lemma 6 shows that equality holds for any two consecutive states, and Lemma 7 shows that it holds over any sequence of operators applicable in the initial state.

**Lemma 6.** *Let $\mathsf{P}$ denote a potential function, let $\mathcal{D}$ denote a disambiguation map, let $\mathsf{Q}$ denote an operator-potential function for $\mathsf{P}$ and $\mathcal{D}$, let $s$ denote a reachable state, and let $o$ denote an operator applicable in $s$. If $|\mathcal{D}(o, V)| = 1$ for every $V \in \text{vars}(\text{eff}(o))$, then $\sum_{f \in s} \mathsf{P}(f) + \mathsf{Q}(o) = \sum_{f \in o[\![s]\!]} \mathsf{P}(f)$.*

*Proof.* Let $A = \bigcup_{V \in \text{vars}(\text{eff}(o))} \mathcal{D}(o, V)$. Since $|\mathcal{D}(o, V)| = 1$ for every $V \in \text{vars}(\text{eff}(o))$, Equation (3) can be rewritten as $\mathsf{Q}(o) = \sum_{f \in \text{eff}(o)} \mathsf{P}(f) - \sum_{f \in A} \mathsf{P}(f)$. And since $s$ is reachable and $o$ is applicable in $s$, it holds that $A \subseteq s$.

Let $B = s \setminus A$. Clearly, $o[\![s]\!] = B \cup \mathrm{eff}(o)$ and $B \cap \mathrm{eff}(o) = \emptyset$. Therefore, $\sum_{f \in s} \mathsf{P}(f) + \mathsf{Q}(o) = \sum_{f \in o[\![s]\!]} \mathsf{P}(f)$ can be rewritten to $\sum_{f \in B} \mathsf{P}(f) + \sum_{f \in A} \mathsf{P}(f) + \mathsf{Q}(o) = \sum_{f \in B} \mathsf{P}(f) + \sum_{f \in \mathrm{eff}(o)} \mathsf{P}(f)$, and further simplified to $\sum_{f \in A} \mathsf{P}(f) + \mathsf{Q}(o) = \sum_{f \in \mathrm{eff}(o)} \mathsf{P}(f)$. Expanding $\mathsf{Q}(o)$ gives us $\sum_{f \in A} \mathsf{P}(f) + \sum_{f \in \mathrm{eff}(o)} \mathsf{P}(f) - \sum_{f \in A} \mathsf{P}(f) = \sum_{f \in \mathrm{eff}(o)} \mathsf{P}(f)$, which concludes the proof. $\square$

**Lemma 7.** *Let* $\mathsf{P}$ *denote a potential function, let* $\mathcal{D}$ *denote a disambiguation map, let* $\mathsf{Q}$ *denote an operator-potential function for* $\mathsf{P}$ *and* $\mathcal{D}$, *let* $\pi = \langle o_1, \ldots, o_n \rangle$ *denote a sequence of operators applicable in* $I$, *and let* $s = \pi[\![I]\!]$. *If* $|\mathcal{D}(o, V)| = 1$ *for every* $o \in \mathcal{O}$ *and every* $V \in \mathrm{vars}(\mathrm{eff}(o))$, *then* $\sum_{f \in I} \mathsf{P}(f) + \sum_{i \in [n]} \mathsf{Q}(o_i) = \sum_{f' \in s} \mathsf{P}(f')$.

*Proof. (By induction)* It clearly holds for an empty sequence $\pi$. Let $s'$ denote a state reachable from $I$ by a sequence $\pi = \langle o_1, \ldots, o_{n-1} \rangle$, and let $o_n \in \mathcal{O}$ denote an operator applicable in $s'$, and let $s = o_n[\![s']\!]$. Now, assume that $\sum_{f \in I} \mathsf{P}(f) + \sum_{i \in [n-1]} \mathsf{Q}(o_i) = \sum_{f' \in s'} \mathsf{P}(f')$, and we need to prove that $\sum_{f \in I} \mathsf{P}(f) + \sum_{i \in [n]} \mathsf{Q}(o_i) = \sum_{f' \in s} \mathsf{P}(f')$. From the assumption, it follows that $\sum_{f \in I} \mathsf{P}(f) + \sum_{i \in [n-1]} \mathsf{Q}(o_i) + \mathsf{Q}(o_n) = \sum_{f' \in s'} \mathsf{P}(f') + \mathsf{Q}(o_n)$, so it is enough to show that $\sum_{f' \in s'} \mathsf{P}(f') + \mathsf{Q}(o_n) = \sum_{f \in s} \mathsf{P}(f)$, which follows from Lemma 6. $\square$

Now, getting to the main result of this section, we formulate an *operator-potential heuristic* and we prove that this heuristic is well-defined and it equals to the corresponding (fact) potential heuristic.

**Definition 8.** Let $\mathsf{Q}$ denote an operator-potential function for $\mathsf{P}$ and $\mathcal{D}$ such that $|\mathcal{D}(o, V)| = 1$ for every $o \in \mathcal{O}$ and every $V \in \mathrm{vars}(\mathrm{eff}(o))$. A **operator-potential heuristic** $h^{\mathsf{Q}} : \mathcal{R} \mapsto \mathbb{R} \cup \{\infty\}$ for $\mathsf{Q}$ is defined as

$$h^{\mathsf{Q}}(s) = \sum_{f \in I} \mathsf{P}(f) + \sum_{i \in [n]} \mathsf{Q}(o_i) \qquad (4)$$

for any sequence of operators $\pi = \langle o_1, \ldots, o_n \rangle$ such that $\pi[\![I]\!] = s$.

**Theorem 9.** *Let* $\mathcal{D}$ *denote a disambiguation map such that* $|\mathcal{D}(o, V)| = 1$ *for every* $o \in \mathcal{O}$ *and every* $V \in \mathrm{vars}(\mathrm{eff}(o))$, *let* $\mathsf{P}$ *denote a potential function, and let* $\mathsf{Q}$ *denote an operator-potential function for* $\mathsf{P}$ *and* $\mathcal{D}$. *Then* $h^{\mathsf{Q}}$ *is well-defined, and* $h^{\mathsf{Q}}(s) = h^{\mathsf{P}}(s)$ *for every reachable state* $s$, *and* $h^{\mathsf{Q}}$ *is admissible (goal-aware, consistent) if* $h^{\mathsf{P}}$ *is admissible (goal-aware, consistent).*

*Proof.* It follows directly from Lemma 7. $\square$

Note that every planning task can be transformed into another task where $|\mathcal{D}(o, V)| = 1$ holds for every operator $o$ and variable $V \in \mathrm{vars}(\mathrm{eff}(o))$. Here, we decided to enforce this property simply by enumerating all possible combinations of facts from all disambiguations $\mathcal{D}(o, V)$ such that $|\mathcal{D}(o, V)| > 1$ for all operators' preconditions. For example, given an operator $o$ with $\mathrm{vars}(\mathrm{eff}(o)) = \{v_1, v_2\}$, and $\mathcal{D}(o, v_1) = \{f_1, f_2\}$ and $\mathcal{D}(o, v_2) = \{f_3, f_4\}$, we replace

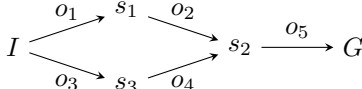

Figure 1: Let $\mathrm{cost}(o_i) = 0$ for all $i \in [4]$, and $\mathrm{cost}(o_5) = 1$, and let $\mathsf{Q}(o_1) = 1$, $\mathsf{Q}(o_2) = 0$, $\mathsf{Q}(o_3) = 0.9$, $\mathsf{Q}(o_4) = 0.1$, and $\mathsf{Q}(o_5) = -1$, and let $h^{\mathsf{Q}}(I) = 0$. A simple example showing inconsistency after rounding operator potentials down to the nearest integers.

the operator $o$ with four new operators $o_1, \ldots, o_4$ with the same effect $\mathrm{eff}(o_1) = \mathrm{eff}(o_2) = \mathrm{eff}(o_3) = \mathrm{eff}(o_4) = \mathrm{eff}(o)$, but we set preconditions to $\mathrm{pre}(o_1) = \mathrm{pre}(o) \cup \{f_1, f_3\}$, $\mathrm{pre}(o_2) = \mathrm{pre}(o) \cup \{f_1, f_4\}$, $\mathrm{pre}(o_3) = \mathrm{pre}(o) \cup \{f_2, f_3\}$, and $\mathrm{pre}(o_4) = \mathrm{pre}(o) \cup \{f_2, f_4\}$. This way, the transformed planning task can grow exponentially in the number of operators. However, in our experiments, we ran out of memory in only one planning task.[2]

## 6 Handling Floating-Point Potentials

Although Theorem 9 identifies conditions under which operator-potential heuristics are consistent and equal to the corresponding potential heuristics, in practice there is an additional complication resulting from the fact that the $\mathsf{Q}(o)$ values are typically represented as floating-point numbers. This means we should not compare the $\mathsf{Q}(o)$ values on equality. Moreover, having floating-point heuristic values is an even larger issue in symbolic search, as states are aggregated based on their $h$-value. If floating-point numbers are used, one could get different BDDs for every state in the search, greatly reducing the effectiveness of symbolic search. Therefore, it is desirable to represent in a single BDD all states whose heuristic values are rounded to the same integer value. Rounding operator potentials down to the nearest integers would resolve this problem and it would keep the heuristic function admissible. Unfortunately, this kind of rounding could make the heuristic inconsistent.

Consider a planning task depicted in Figure 1. Clearly, the operator-potential heuristic $h^{\mathsf{Q}}$ is both admissible and consistent. Now, let $\hat{\mathsf{Q}}$ denote an operator potential function obtained by rounding down $\mathsf{Q}$ to the nearest integers, i.e., $\hat{\mathsf{Q}}(o_1) = 1$, $\hat{\mathsf{Q}}(o_2) = 0$, $\hat{\mathsf{Q}}(o_3) = 0$, $\hat{\mathsf{Q}}(o_4) = 0$, and $\hat{\mathsf{Q}}(o_5) = -1$. The sum $\sum_{f \in I} \mathsf{P}(f) + \sum_{i \in [n]} \mathsf{Q}(o_i)$ (cf. Definition 8) provides an admissible estimate, because rounding down can make the sum only smaller. However, rounding down can also make this estimate path-dependent, i.e., we can obtain different values for a state depending on the path by which we reached the state, and inconsistent.

Consider the states $s_1$ and $s_2$, and operator sequences $\pi = \langle o_1 \rangle$ and $\pi' = \langle o_3, o_4 \rangle$ from Figure 1. Since the heuristic value for the initial state is zero, the inequality $h^{\mathsf{Q}}(s_1) - h^{\mathsf{Q}}(s_2) \leq \mathrm{cost}(o_2)$ holds, because $h^{\mathsf{Q}}(s_1) = 1$, $h^{\mathsf{Q}}(s_2) = 1$,

[2] A possibility for future work is the use of transition normal form (Pommerening and Helmert 2015) which is polynomial, but introduces a set of auxiliary operators, and requires a transformation of the resulting plan back to the original planning task.

and $\text{cost}(o_2) = 0$. But after rounding, we get $\hat{\mathtt{Q}}(o_1) = 1$ and $\hat{\mathtt{Q}}(o_3) + \hat{\mathtt{Q}}(o_4) = 0$ resulting in a higher estimate for $s_1$ than for $s_2$ using $\hat{\mathtt{Q}}$.

We resolve this issue by encoding the rounding directly into the (integer) linear program expressing the potentials. Since the operator potential is just a left hand side of Eq. (2) with the opposite sign, we can create a new integer variable for each operator potential and add a new constraint

$$\mathtt{Q}(o) = \sum_{f \in \text{eff}(o)} \mathtt{P}(f) - \sum_{V \in \text{vars}(\text{eff}(o))} \max_{f \in \mathcal{D}(o,V)} \mathtt{P}(f) \quad (5)$$

for each operator $o \in \mathcal{O}$. This way, (fact) potentials $\mathtt{P}$ can still be real-valued, but the operator potentials will have integer values. Therefore a proper rounding of operator potentials will be done by the ILP solver, and we can compare them on equality without running into problems with floating-point numbers.

## 7 Symbolic Search with Potential Heuristics

Using potential heuristics in BDDA$^*$ is not straightforward, as the standard way of evaluating the heuristics by constructing a BDD $S_h$ representing all states with a heuristic value equal to $h$ may not always be feasible. The naive approach would be to enumerate all possible sub-sets of features whose potentials add up-to $h$. However, this requires enumerating exponentially many sub-sets and it may easily result in an exponentially large BDD. We overcome this difficulty by using operator-potential heuristics instead.

To do so, we use GHSETA$^*$ (Jensen, Veloso, and Bryant 2008), a symbolic heuristic search algorithm which partitions the TRs not only by the cost of the corresponding operators, but also by the change of the heuristic value they induce. That is, instead of creating a TR $T_c$ for all operators $o$ having $\text{cost}(o) = c$, we create a TR $T_{c,q}$ representing all operators $o$ such that $\text{cost}(o) = c$ and $\mathtt{Q}(o) = q$. For the initial state, the $g$-value is set to zero, and the $h$-value is set to $\sum_{f \in I} \mathtt{P}(f)$. For all subsequent states $S_{g,h}$ expanded by the TR $T_{c,q}$, the $g$-value and $h$-value of the resulting state $S'_{g',h'} = \text{image}(S_{g,h}, T_{c,q})$ is set to $g' = g + c$ and $h' = h + q$, respectively.

The pseudocode for the algorithm using a *consistent* operator-potential heuristic is encapsulated in Algorithm 1. On lines 1 and 2, the TRs corresponding to all unique pairs of operator costs and $\mathtt{Q}(o)$ values are constructed. The heuristic value for the initial state is computed on line 3. The open list of sets of states (represented by BDDs) ordered by $f = g + h$ values is initialized with the initial state on line 4 and 5. On line 6, a BDD representing all closed states is constructed. The while-cycle on lines 7–15 is an A$^*$ algorithm adapted to the symbolic search. On line 8, we extract the set of states with the lowest $f$-value from the open list (the function PopMin()) and remove all closed states from this set. If a goal state is reached (line 9 and 10), an optimal plan is extracted and returned (for details see (Torralba et al. 2017)). If the current set of states $S_{g,h}$ does not contain a goal state, then all these states are added to the set of closed states (line 11). On lines 12–15, all operators are applied and the resulting states that were not closed yet are assigned the

---

**Algorithm 1:** Symbolic forward A$^\star$ with consistent operator-potential heuristic.

**Input:** A planning task $\Pi$, an operator potential function $\mathtt{Q}$ for $\mathtt{P}$ and $\mathcal{D}$.
**Output:** An optimal plan or "unsolvable".

1 **for each** $c, q \in \{\text{cost}(o), \mathtt{Q}(o) \mid o \in \mathcal{O}\}$ **do**
2 $\quad$ Construct $T_{c,q}$ from $\{o \in \mathcal{O} \mid \text{cost}(o) = c, \mathtt{Q}(o) = q\}$;
3 $h_I \leftarrow \sum_{f \in I} \mathtt{P}(f)$;
4 $S_{0,h_I} \leftarrow \{I\}$;
5 open $\leftarrow \{S_{0,h_I}\}$;
6 closed $\leftarrow \emptyset$;
7 **while** *open* $\neq \emptyset$ **do**
8 $\quad S_{g,h} \leftarrow \text{PopMin}(\text{open}) \setminus \text{closed}$;
9 $\quad$ **if** $S_{g,h}$ *contains a goal state* **then**
10 $\quad\quad$ **return** ExtractPlan($S_{g,h}$);
11 $\quad$ closed $\leftarrow$ closed $\cup\ S_{g,h}$;
12 $\quad$ **for each** $T_{c,q}$ **do**
13 $\quad\quad S_{g+c,h+q} \leftarrow \text{image}(S_{g,h}, T_{c,q}) \setminus \text{closed}$;
14 $\quad\quad$ **if** $S_{g+c,h+q} \neq \emptyset$ **then**
15 $\quad\quad\quad$ InsertOrUpdate(open, $S_{g+c,h+q}$);
16 **return** "unsolvable";

---

correct $g$ and $h$ values and either inserted into the open list (if there is no $S_{g+c,h+q}$ in the open list), or the set of states in the open list is extended with the new set of states.

Note that we need the operator-potential heuristic to be consistent in order to avoid re-opening states (cf. lines 8 and 13). This also means that if the computation of consistent operator-potentials require the transformation of the planning task described in Section 5, then the same transformed task must be used also for the symbolic search.

## 8 Experimental Evaluation

We implemented our search algorithm in C. Operators and facts are pruned with the $h^2$ heuristic in forward and backward direction (Alcázar and Torralba 2015), and the translation from PDDL to FDR uses the inference of mutex groups proposed by Fišer (2020). We used all planning domains from the optimal track of International Planning Competitions (IPCs) from 1998 to 2018 excluding the ones containing conditional effects after translation. We merged, for each domain, all benchmark suites across different IPCs. This leaves 48 domains overall.

The experiments were run on a cluster of computing nodes with Intel Xeon Scalable Gold 6146 processors and CPLEX solver v12.9 used for the computation of potentials. The time and memory limits were set to 30 minutes and 8 GB, respectively. All variants use a time limit of 30 seconds for applying mutexes on the BDD representing goal states and 10 seconds time limit for merging BDDs representing transition relations (Torralba et al. 2017).

We evaluated GHSETA$^*$ with the following variants of operator-potential heuristics:

- I: optimization for the initial state (Pommerening et al. 2015), i.e., maximize the heuristic value for the initial state.

- A+I: optimization for all syntactic states with added constraint on the initial state (Seipp, Pommerening, and

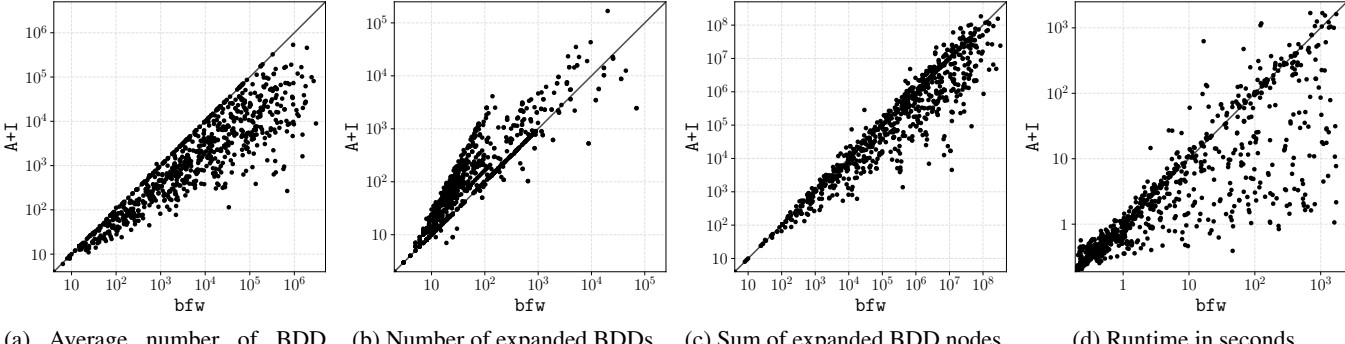

(a) Average number of BDD nodes per expanded BDD.

(b) Number of expanded BDDs.

(c) Sum of expanded BDD nodes.

(d) Runtime in seconds.

Figure 2: Comparison of symbolic forward uniform-cost search (`bfw`) against GHSETA$^*$ with the best-performing variant of the potential heuristic (`A+I`) on tasks that were solved by both variants.

Helmert 2015; Fišer, Horčík, and Komenda 2020), i.e., maximize the heuristic value for an average (syntactic) state while enforcing the maximum heuristic value for the initial state.

- $S_{1k}$+I: optimization for an average over 1 000 random states with added constraint on the initial state (Seipp, Pommerening, and Helmert 2015; Fišer, Horčík, and Komenda 2020), i.e., sample 1 000 states using random walks and maximize the heuristic value for the average state over the sampled states while enforcing the maximum heuristic value for the initial state.

- $M_2$+I: optimization for all reachable states approximated with mutexes with added constraint for the initial state (Fišer, Horčík, and Komenda 2020), i.e., approximate the number of reachable states by enumerating all possible pairs of facts, and then use mutexes to approximate the number of reachable states containing each fixed pair of facts.

We compare these to symbolic uniform-cost search using forward (`bfw`) and bidirectional search (`bbi`). Furthermore, we compare to other state-of-the-art planners. We ran A$^*$ with the LM-Cut (`lmc`) heuristic (Helmert and Domshlak 2009), with the merge-and-shrink (`ms`) heuristic with SCC-DFP merge strategy and non-greedy bisimulation shrink strategy (Helmert et al. 2014; Sievers, Wehrle, and Helmert 2016), and with the potential heuristic (`pot`$_{A+I}$) optimized for all syntactic states with added constraint for the initial state (Fišer, Horčík, and Komenda 2020) (i.e., a variant of `A+I` for A$^*$). We further compare to two of the best-performing non-portfolio planners from IPC 2018: Complementary2 (`comp2`) (Franco et al. 2017; Franco, Lelis, and Barley 2018), and Scorpion (`scrp`) (Seipp 2018; Seipp and Helmert 2018).

Table 1 shows the coverage comparison across all planners, in terms of total coverage, and in terms of the number of domains in which each planner is superior to others. Table 2 gives detailed per-domain coverage results.

| | scrp | A+I | $M_2$+I | comp2 | $S_{1k}$+I | bbi | pot$_{A+I}$ | I | bfw | ms | lmc | tot |
|---|---|---|---|---|---|---|---|---|---|---|---|---|
| scrp | – | **20** | **22** | **19** | **24** | **27** | **26** | **29** | **31** | **33** | **36** | 1 112 |
| A+I | 14 | – | **9** | **16** | **20** | **21** | **24** | **31** | **29** | **33** | **34** | 1 109 |
| $M_2$+I | 14 | 1 | – | **14** | **18** | **20** | **24** | **27** | **27** | **32** | **34** | 1 097 |
| comp2 | 16 | 16 | **18** | – | **24** | **19** | **26** | **26** | **30** | **32** | **33** | 1 091 |
| $S_{1k}$+I | 12 | 3 | 8 | 16 | – | **19** | **22** | **29** | **27** | **31** | **30** | 1 081 |
| bbi | 14 | 14 | 14 | 12 | 17 | – | **24** | **23** | **27** | **26** | **26** | 1 010 |
| pot$_{A+I}$ | 5 | 10 | 12 | 13 | 14 | 17 | – | **21** | **23** | 17 | **24** | 1 002 |
| I | 9 | 1 | 3 | 9 | 4 | 15 | 14 | – | **20** | 17 | **21** | 989 |
| bfw | 10 | 4 | 4 | 6 | 8 | 6 | 17 | 10 | – | **19** | 16 | 933 |
| ms | 2 | 6 | 7 | 7 | 10 | 14 | 9 | 15 | **22** | – | **19** | 933 |
| lmc | 2 | 5 | 6 | 6 | 9 | 13 | 14 | 17 | **23** | 18 | – | 911 |

Table 1: Summary of domain coverage. A value in row $x$ and column $y$ is the number of domains where $x$ solved more tasks than $y$, it is bold if higher than the value in row $y$ and column $x$. "tot" shows overall number of solved tasks. Highlighted rows correspond to GHSETA$^*$ with operator-potential heuristics.

## 8.1 Operator Potentials in Symbolic Search

Consider first the comparison of symbolic potential variants against the baseline forward search without heuristics (`bfw`). This clearly demonstrates that potential heuristics are beneficial for the performance of symbolic search over a wide range of different domains. The best variant of GHSETA$^*$ with an operator-potential heuristic (`A+I`) solves 176 more tasks than the baseline (`bfw`). It increases coverage on 29 different domains, and it is detrimental in only 4 domains. Among the different variants of potential heuristics, we observe that the optimization criteria can have a significant impact on performance. The best variant is `A+I`, closely followed by $S_{1k}$+I and $M_2$+I, and significantly better than I. These results are well in line with the results of the same potential heuristics in explicit search (Fišer, Horčík, and Komenda 2020).

For a heuristic to be beneficial in symbolic search, it is required that sets of states with the same $g$ and $h$ are efficiently

| Domain | GHSETA$^*$ + potentials | | | | bfw | bbi | lmc | ms | pot$_{A+I}$ | comp2 | scrp |
|---|---|---|---|---|---|---|---|---|---|---|---|
| | I | A+I | S$_{1k}$+I | M$_2$+I | | | | | | | |
| agricola18 (20) | 15 | 19 | 17 | **20** | 19 | **20** | 0 | 4 | 3 | 10 | 6 |
| airport04 (50) | 26 | 26 | 26 | 24 | 24 | 26 | 30 | 21 | 36 | 28 | **40** |
| barman11/14 (34) | 11 | 14 | 14 | 14 | 15 | **16** | 10 | 11 | 11 | 15 | 11 |
| blocks00 (35) | 21 | 31 | 29 | 31 | 22 | **33** | 28 | 21 | 28 | 31 | 28 |
| caldera18 (20) | 16 | 17 | 17 | 16 | **18** | **18** | 12 | 12 | 12 | 15 | 13 |
| cavediving14 (20) | 7 | 7 | 7 | 7 | 7 | **8** | 7 | 7 | 7 | 7 | 7 |
| childsnack14 (20) | 4 | **5** | **5** | **5** | **5** | **5** | 0 | 0 | 0 | 2 | 0 |
| data-network18 (20) | 8 | 13 | 9 | 13 | 11 | 13 | 12 | 13 | 9 | 13 | **14** |
| depot02 (22) | 7 | 11 | 10 | 11 | 6 | 8 | 7 | 11 | 11 | 8 | **14** |
| driverlog02 (20) | 13 | 14 | 13 | 14 | 11 | 14 | 14 | 13 | 13 | **15** | **15** |
| elevators08/11 (50) | 35 | 35 | 35 | 35 | 35 | 43 | 40 | 35 | 31 | **44** | **44** |
| floortile11/14 (40) | 17 | 18 | 17 | 17 | 17 | **34** | **34** | 16 | 11 | **34** | 16 |
| freecell00 (80) | 47 | 68 | 68 | 67 | 20 | 27 | 15 | 62 | **72** | 32 | **72** |
| ged14 (20) | 15 | 15 | 16 | 15 | 15 | **20** | 19 | 19 | 15 | **20** | **20** |
| gripper98 (20) | **20** | **20** | **20** | **20** | **20** | **20** | 8 | 9 | 8 | **20** | 8 |
| hiking14 (20) | 14 | 16 | 15 | 16 | 16 | 18 | 11 | 14 | 14 | **20** | 15 |
| logistics98/00 (63) | 23 | 28 | 27 | 28 | 21 | 25 | 26 | 25 | 24 | 28 | **37** |
| maintenance14 (5) | **5** | **5** | **5** | **5** | **5** | **5** | **5** | **5** | **5** | **5** | **5** |
| movie98 (30) | **30** | **30** | **30** | **30** | **30** | **30** | **30** | **30** | **30** | **30** | **30** |
| mprime98 (35) | 27 | **31** | 28 | **31** | 27 | 16 | 25 | 24 | 24 | 24 | **31** |
| mystery98 (30) | 15 | **19** | **19** | **19** | 15 | 10 | 17 | 17 | 18 | 15 | **19** |
| nomystery11 (20) | 13 | 19 | 16 | 19 | 12 | 16 | 16 | 14 | 14 | **20** | **20** |
| openstacks06/08/11/14 (100) | **91** | **91** | **91** | **91** | 85 | 86 | 51 | 56 | 57 | 74 | 55 |
| organic-synthesis18 (20) | **10** | **10** | **10** | **10** | **10** | **10** | **10** | 7 | **10** | **10** | **10** |
| parcprinter08/11 (50) | 44 | 48 | 47 | 47 | 40 | 37 | 41 | 44 | 48 | 43 | **50** |
| parking11/14 (40) | 0 | 13 | 12 | 13 | 0 | 6 | 8 | 2 | **16** | 5 | **16** |
| pathways06 (30) | **5** | **5** | **5** | **5** | **5** | **5** | **5** | 4 | 4 | **5** | **5** |
| pegsol08/11 (50) | 48 | 48 | 46 | 48 | 46 | 48 | 48 | 48 | 48 | 48 | **50** |
| petri-net-alignment18 (20) | 9 | 10 | 10 | 10 | 16 | **19** | 9 | 0 | 13 | **19** | 0 |
| pipesworld-notankage04 (50) | 22 | 26 | 26 | 24 | 17 | 17 | 18 | 23 | **30** | 25 | 26 |
| pipesworld-tankage04 (50) | 18 | **19** | 18 | **19** | 17 | 15 | 13 | 16 | **19** | **19** | 18 |
| psr-small04 (50) | **50** | **50** | **50** | **50** | **50** | **50** | 49 | **50** | **50** | **50** | **50** |
| rovers06 (40) | 13 | **14** | **14** | **14** | **14** | **14** | 9 | 8 | 8 | 13 | 10 |
| satellite02 (36) | 7 | 10 | 9 | 10 | 7 | **12** | 9 | 6 | 6 | 10 | 10 |
| scanalyzer08/11 (50) | 23 | 21 | 19 | 21 | 21 | 21 | 31 | 23 | 23 | 21 | **33** |
| snake18 (20) | 10 | 10 | 11 | 8 | 7 | 0 | 7 | **15** | **15** | 13 | **15** |
| sokoban08/11 (50) | 48 | **50** | **50** | 48 | 48 | 48 | **50** | **50** | **50** | 48 | **50** |
| spider18 (20) | 12 | 13 | 11 | 12 | 7 | 7 | 11 | 12 | **16** | 13 | **16** |
| storage06 (30) | 15 | **16** | **16** | **16** | 15 | 15 | 15 | 15 | **16** | 15 | **16** |
| termes18 (20) | 12 | 12 | 12 | 12 | 12 | **18** | 6 | 14 | 12 | 16 | 14 |
| tetris14 (17) | 13 | 16 | 16 | 16 | 9 | 10 | 9 | 13 | **17** | 13 | 13 |
| tidybot11/14 (40) | 29 | 33 | 32 | 33 | 25 | 12 | 30 | 30 | 32 | **39** | 35 |
| tpp06 (30) | 11 | 12 | 12 | 12 | 8 | 8 | 7 | 8 | 8 | **15** | 8 |
| transport08/11/14 (70) | 23 | 24 | 24 | 24 | 26 | 34 | 23 | 27 | 24 | 33 | **38** |
| trucks06 (30) | **16** | **16** | 15 | **16** | 13 | 14 | 13 | 10 | 14 | 14 | **16** |
| visitall11/14 (40) | 22 | 22 | 22 | 22 | 17 | 19 | 18 | 29 | 30 | **33** | 30 |
| woodworking08/11 (50) | 38 | 46 | 48 | 46 | 38 | 48 | 42 | 29 | 29 | 48 | **50** |
| zenotravel02 (20) | 11 | **13** | 12 | **13** | 9 | 12 | **13** | 11 | 11 | **13** | **13** |
| $\Sigma$ (1697) | 989 | 1 109 | 1 081 | 1 097 | 933 | 1 010 | 911 | 933 | 1 002 | 1 091 | **1 112** |

Table 2: Number of solved tasks per domain and overall. The best results in each row are highlighted in bold.

represented with BDDs. The positive coverage results from Table 1 suggest that this is indeed the case for the operator-potential heuristics. To confirm this, Figure 2 compares the performance of the baseline, symbolic search without any heuristics (bfw), against the best configuration of our symbolic search with potential heuristics (A+I), according to different metrics.

First, we observe that the average number of BDD nodes per expanded BDD was almost always lower (Figure 2a). This means that indeed, the partitioning induced by operator-potential heuristics is often beneficial, resulting on sets of states during the search that have a concise BDD representation. This property, however, is not guaranteed by the method. In particular, the average number of BDD nodes per expanded BDD was higher for bfw in only ten tasks, though only by a small margin. On the other hand, the number of expanded BDDs is almost always increased (Figure 2b), which is not surprising, as sets of states during the search are not only partitioned by $g$ value, but also by $h$ value.

Most remarkably, the number of BDD nodes from all expanded BDDs (sets of states) often decreased with potential heuristics (Figure 2c). This confirms that these heuristics are not only informative for explicit-state search, avoiding the expansion of certain states, but also are beneficial in symbolic search by inducing a good BDD partitioning. This contrasts with previous results on very informative heuristics ($\frac{1}{2}h^*$, $\frac{3}{4}h^*$), that despite their accuracy are often detrimental for the performance of symbolic search (Speck, Geißer, and

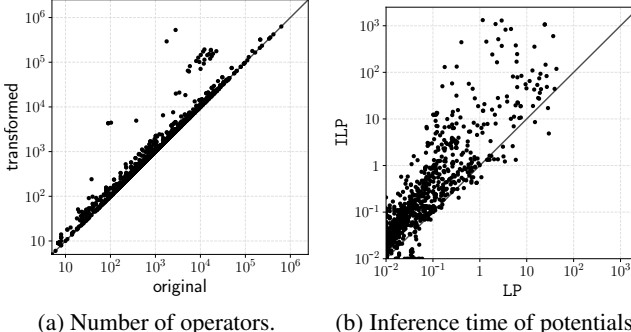

(a) Number of operators.    (b) Inference time of potentials.

Figure 3: Comparison of (a) the number of operators before and after the transformation, and (b) a time (in seconds) spent in a computation of potentials for the original formulation (LP) and with added constraints on integer operator potentials (ILP).

Mattmüller 2020).

Furthermore, the runtime of the planner is often decreased (Figure 2d), confirming that partitioning the TRs according to the operator potentials is a very effective way of evaluating the heuristic in symbolic search. Therefore, not only potential heuristics can be informative for symbolic search, but they can be efficiently evaluated also. The increase in the runtime for some of the planning tasks is often due to increased computational effort in the inference of integer operator potentials, as we analyze in detail next.

Compared against symbolic bidirectional uniform cost search (bbi), which has state-of-the-art performance in symbolic search planning (Torralba et al. 2014; Torralba, Linares López, and Borrajo 2016), A+I solves 99 more tasks. The two algorithms are still quite complementary though, with A+I being superior in 21 domains and bbi in 14. This suggests that there is still potential to further improve results by integrating operator potential heuristics in symbolic bidirectional heuristic search.

## 8.2 Comparison Against Explicit-State Search with Potential Heuristics

Compared to the potential heuristics in explicit-state search, A+I solves 107 instances more than $pot_{A+I}$. Moreover, there are only 10 domains where using symbolic search is detrimental, compared to 24 domains where it is beneficial.

Note that these two configurations are using the same optimization criteria to compute the potentials. However, as explained in Section 6, in order to obtain consistent operator-potential heuristics, we must (1) split the operators so that all variables mentioned in the effect appear in the preconditions; and (2) use ILP to ensure that operator potentials have an integer value. This sometimes has an overhead, as illustrated by Figure 3, which makes the improvements in coverage even more remarkable.

Figure 3a compares the number of operators in the original planning task and in the corresponding task where all variables affected by operators are explicitly defined in preconditions, i.e., in the planning task transformed using the method described in Section 6. It shows that in most cases the transformation does not significantly increase the size of the task. However, there are few domains where the number of operators increases significantly. In particular, the transformed tasks are more than five times bigger in two domains: agricola18 and caldera18. Note that these are two out of five domains where our configurations had lower coverage than the baseline (see Table 2), suggesting that the few negative results are not due to potential heuristics being detrimental but rather due to the overhead of the pre-processing stage.

Figure 3b shows that the inference of the integer operator potentials using ILP takes often significantly longer than finding (fact) potentials with LP. However, it turns out there is usually enough time left for the search part.

## 8.3 Comparison Against State of the Art

Compare finally the performance of our best-performing variants (A+I, $M_2$+I, and $S_{1k}$+I) against the unrelated approaches lmc, ms, comp2, and scrp. Our planners clearly beat lmc and ms in terms of overall coverage and frequency of per-domain superiority. The state-of-the-art planners comp2 and scrp are roughly on par in overall coverage. In terms of individual domains, the clear conclusion is that our new techniques are highly complementary to the previous state of the art: A+I, $M_2$+I, and $S_{1k}$+I outperform comp2 and scrp in 12–16 domains.

## 9 Conclusion

While heuristic search and symbolic search are both contenders for the throne in optimal planning, and their combination is a natural and promising avenue, the results with that combination have thus far been disappointing. As we show, this picture changes dramatically when leveraging the fact that potential heuristics can be viewed as potentials over operators, which enables their smooth integration into symbolic search. We have shown that and how this can be done, in particular while retaining consistency. Our empirical results show that this boosts the performance of optimal symbolic planning, which is now on par with the best heuristic search based optimal planners.

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
