# OpenReview forum: "Introducing Operator-Potential Heuristics for Symbolic Search"
_icaps-conference.org/ICAPS/2021/Workshop/HSDIP — HSDIP 2021_

### Official Review · AnonReviewer1 · 2021-05-26
**This paper was an excellent read**

**Confidence:** 4
**Overall Score:** Strong Accept

**Review:**

Summary of the paper:

The paper introduces a new state of the art in symbolic search planning by
integrating potential heuristics in symbolic search. The key insight is that
potential heuristics can be encoded as operator potentials, which allows to
integrate the heuristic directly into the transition relation function, avoiding
a large fractioning of the BDDs that other symbolic heuristics often induce.

Summary of the review:

This paper is well-written and was a pleasure to read. The formal background of
the idea is rigorously defined, well motivated and the proofs are correct and
easy to follow. The arguments and reasoning of the paper are clear and
reasonable, at least for readers familiar with potential heuristics and symbolic
search. An extensive empirical evaluation shows that this new approach
outperforms the previous state of the art in symbolic search and almost presents
a new state of the art in optimal classical planning in general. Recent results
for symbolic search are also taken into consideration and discussed in detail.
All in all, this paper has the quality of submissions to major A* AI conferences
and should clearly be included in the workshop.

I have a few comments and suggestions on some of the presentation, and a bit
more detailed feedback regarding atomic disambiguations and the relation to the
transitional normal form. I hope the authors take time during the discussion
period to discuss some of my points.


Major feedback:

- I have some issues with the wording of the discussion of the example for
floating-point problems starting on line 361. In particular, $\hat{Q}$ is denoted
as an operator potential function, but Definition 5 does not apply to $\hat{Q}$.
Similarly, $h^{\hat{Q}}$ can not be denoted as an operator-potential heuristic,
because it simply is not one, given Definition 8 (which is further backed up by
Theorem 9, because if it was an operator-potential heuristic it would be
consistent!). In particular $h^{\hat{Q}}(s_2)$ is not well-defined, because the
sequence $o_1,o_2$ has a different Q-sum than the sequence $o_3,o_4$, but Definition 8
requires that the h-value is equal for any sequence of operators.

 I think the example is great to show the underlying problem, but given that the
previous section rigorously defined what an operator-potential heuristic is the
wording of Section 6 should be adapated.

- The discussion of Figure 2 is highly interesting and I commend the paper for
going into such detail. However, I want to highlight that the paper never really
introduces to the reader the concept of BDD nodes and in fact I think the paper
never mentions that the size of a BDD is the number of nodes or what an
'efficient representation' means. Readers not familiar with BDDs might thus
complain that the paper is not self-contained. I suggest to add one or two
sentences on the symbolic search background to make this more clear.

Detailed discussion / questions:

- It took me some time to realize that Def. 3 (i) implies that for variables
that appear in the precondition and the effect (i.e. $V \in vars(pre(o)) \cap vars(eff(o)))$
any disambiguation $F$ has to include $\langle V, pre(o)[V] \rangle$, since otherwise we can not have
$pre(o) \subseteq s$ and $F \cap s \neq \emptyset$. More importantly, if we restrict
the size of $F$ to 1 this implies that$ \mathcal{D}(o,V) = pre(o)[V]$ for
$V \in vars(pre(o)) \cap vars(eff(o)))$, and therefore for operators where
precondition and effect share the same variables the operator-potential
function simply becomes the sum of potential of effect facts minus the sum of
potentials of precondition facts. Is this correct, or did I miss / misunderstand
something?

 If this is correct, then this would imply that the proofs would be even simpler
assuming transition normal form (TNF), because TNF guarantees that  $vars(pre(o))
= vars(eff(p))$. But then I think Eq. 2 of Theorem 4 would also be equivalent to
Def. 4 of Pommerening et al. (2015), i.e. to the original definition of
potential heuristics. This raises the following question: if we use TNF, do we
gain anything from using a disambiguation mapping? Clearly such a discussion is
out of scope of the current paper (especially if the paper is planned to be
submitted to AAAI), but perhaps we can use the open discussion period to discuss
this in more detail.

- That being said, as far as I see the reason to restrict the disambiguation size
to 1 is the argument of the proof of Lemma 6 in line 284/275 (please correct me
if I am wrong). I see that the proof would not hold for sizes > 1, but I don't
see that the general statement of Theorem 9 breaks for arbitrary disambiguation
sizes. Is the reason that you restrict the size to 1 that this makes proving
the theorem simpler, or is there a counter-example that shows that the theorem
does not hold for arbitrary size? In particular, I wonder whether it could be
possible to allow for higher-dimensional potential heuristics (Pommerening,
Helmert, Bonet 2017 (AAAI)) instead of atomic potentials.


Minor comments:

- Line 78: heuristic -> heuristics

- Line 110: is there a particular reason that you use $\mathbb{Z}^+$ instead of
$\mathbb{N}$?

- Line 241/242: minimally misleading: Definition 1 only describes atomic
potential heuristics, but higher-dimensional potential heuristics have also been
used.

- Line 315: 'An' operator-potential heuristic

- For the first part of the proof of Lemma 6: "since s is reachable and o is
applicable in s, it holds that $ A \subseteq s$", I think it might help to add
that 'it holds' also follows from the definition of disambiguation (i.e. an
atomic disambiguation f is contained in all reachable states where $p \subseteq
s$).

- Line 529: I suppose 'was higher' should be 'was lower'? If not, then I don't
understand the argument here

- It is not clear from Section 8 whether the task transformation to guarantee
atomic disambiguations is only done for the precomputation phase to obtain the
potentials, or whether search itself is performed on the transformed task.

---

> ### Author Response · Authors · 2021-06-02
> **Response**
>
> Thanks for your review.
>
> 1. We think your comment on \hat{Q} is spot-on -- that was our oversight.
> If accepted, we'll make clear in the camera-ready that h^\hat{Q} becomes
> a path-dependent heuristic that is not necessarily consistent. We think, we can
> do that without introducing any new notation, because the point we are making
> is that the sums over different paths in Fig. 1 give different numbers.
>
> 2. Regarding Fig. 2: We'll introduce BDD nodes, probably in Section 3. We
> have enough space left.
>
> 3. Regarding disambiguations of variables in vars(pre(o)) \cap vars(eff(o)): You
> understood correctly. This is already discussed in the referred paper by Fiser et
> al. (2020) on potential heuristics with disambiguations. (We will, however, add a couple
> of sentences that will make this part easier to understand.)  This goes also to the TNF
> representation -- in Fiser et al. (2020), there is a section dedicated to TNF that should
> clarify how the disambiguations are connected to TNFs. In short, application of
> (multi-fact) disambiguation to potential heuristics can be understood as a construction of
> the equivalent TNF representation that represents the task more concisely than the
> "original" TNF. The potentials inferred on such TNF are then equivalent to the ones
> obtained with the LP modified with the disambiguation.
>
> 4. Theorem 9 doesn't work for disambiguations of arbitrary size. Consider
> a binary variable V with values 0/1 and an operator o with a precondition on
> some other variable and the effect V=0. If V=0 and V=1 have different
> potentials, say P(V=0) = 1 and P(V=1) = 2, then Q(o) = -1. But, depending
> on the state where o is applied, the actual change of potential heuristic may
> be either zero or -1. This would lead to a path-dependent heuristic, i.e.,
> h^P(s) = h^Q(s) does not necessarily hold anymore.
>
> However, we think we can make the symbolic search work even with the
> path-dependent variant of the operator-potential heuristic, which we plan to
> address in the next iteration of the paper.
>
> 5. We think this approach should, in principle, work also for high-dimensional
> potentials. However, we are not sure about the use of disambiguations -- this
> will require further investigation.
>
> 6. Regarding your last question, the transformation of the task must be used
> not only for the operator-potentials but also for the whole search. Consider
> again the operator o from 4.: In the transformed task, there would be two
> variants of o: o' would have V=0 as a precondition, and o'' would have V=1 as
> a precondition, and therefore Q(o') = 0 and Q(o'') = -1. So, o' and o'' will
> be part of different transition relation BDDs.
> We'll make that clear in the camera-ready.

---

### Official Review · AnonReviewer2 · 2021-05-26

**Confidence:** 4
**Overall Score:** Strong Accept

**Review:**

In this paper, an approach to classical planning that combines symbolic search and heuristic search is presented. While several approaches for symbolic heuristic search have been presented in the past, they failed to consistently outperform symbolic blind search. The approach in this paper utilizes potential heuristics for operators to efficiently perform symbolic heuristic search. The theoretical basis for this approach is provided, and a convincing empirical evaluation is performed.

The paper overall is well written and all concepts and ideas are nicely motivated and easy to follow. I enjoyed reading the paper and think it is a great contribution to the workshop.

I have a few comments and questions, mostly related to the importance of partitioning BDDs in general, versus the particular partitioning induced by the potential of operators.

1. Impact of partitioning vs. impact of heuristic estimates:
For a heuristic to perform well in the context of symbolic heuristic search, a good partitioning of the states is required in the sense that the resulting BDDs can be represented compactly and efficiently (Property 2). That partitioning can provide a large performance improvement in symbolic search is well known and is also stated in the paper. The results of the paper show that such good partitionings are possible by the described approach. But at the same time, some states do not need to be expanded due to the use of the heuristic function. I wonder how big the influence of the latter is on the search performance. In other words, is the performance improvement mainly due to the partitioning and the heuristic estimates play a minor role or do these estimates and the pruned states really have an impact. This may be tested by running the search and "ignoring" the heuristic values and continuing after finding a goal state and expanding the sets of states until the g-values have the value as the found goal/solution.


2. Empirical evaluation:
- Is the implementation based on the symBA planner? If not, it would be interesting to see how the bidirectional blind search configuration differs from the base version of symBA on IPC 2018. I think a comparison with symBA (with all its extensions) would be very interesting, using the same parameters like variable order, mutexes, etc.
- It seems that all operators with cost c have been merged in the symbolic blind search configurations, which to my knowledge has proven to be infeasible in many instances (Torralba et al. 2013). Again, the question is whether the use of reasonable partitioning of transition relations (e.g., an upper bound on BDD nodes per TR) is the key factor in improving performance, or whether the heuristic estimates and induced partitioning of transition relations and state sets are the critical part. Are there any findings on this?
- It would be interesting to know what strategy was used in the bidirectional search to determine which search front to expand? I think it would be good to publish the code, so that these details could be checked by an interested reader.


3.  Finally, I wonder if the authors have looked more closely at using their approach for bidirectional search. With the TNF mentioned, it might be possible to compute potentials for the backward search direction as well.

Overall, the paper is an important contribution to symbolic heuristic search and fits perfectly to the workshop, so I (strongly) recommend its acceptance.



Minors:
- Preliminaries: N_0 instead of Z^+_0 ?
- It may be beneficial to briefly explain what a BDD is and introduce the concept of the size/node of a BDD.

---

> ### Author Response · Authors · 2021-06-02
> **Response**
>
> Thanks for your review.
>
> We plan to address all of your questions in the next iteration of the paper
> which we intend to submit to a conference:
>
> 1. Regarding the impact of partitioning vs. heuristic estimates: We already
> have some preliminary experimental results where we ran blind search, but we
> kept the same partitioning as for the potential heuristics and we didn't merge
> the BDDs with the same f-values (so that we don't lose the partitioning).
> The performance with such configuration was often worse than with the original
> partitioning (without potentials), so we think the partitioning itself is not
> enough, but the information provided by the heuristics is crucial in order to
> achieve the results as presented in the paper.
>
> However, we find it striking that the partitioning using potential heuristics
> almost always produces a good partitioning for symbolic search and we would be
> surprised if that was by a lucky coincidence. Analyzing this behaviour will
> require more fine-grained approach and, currently, we are not even sure how to
> do that. We hope a discussion with other researchers at HSDIP will help us
> find a way to analyze this behaviour.
>
> 2. We implemented the planner from scratch according to Torralba et al.'s
> paper, and we made sure the results for blind forward and bi-directional
> search are on-par with the original implementation used as a baseline in IPC 2018. That said, the main difference is that the original implementation uses
> a time limit for image() operation effectivelly allowing to switch-off
> backward search if it takes too long. In our implementation, we don't use this
> procedure at all. Other than that, our implementation uses mutexes,
> partitioning of transition relation and states into multiple BDDs, and the
> same strategy for switching between forward and backward search as described
> in the Torralba's paper (i.e., the time needed for image() operation is based
> on the size of the BDD and the time needed in the previous step).
> We definitely intend to publish the code once we got accepted to a conference.
>
> 3. Yes, we looked into the bi-directional search, and it definitely can be
> done. But after spending some time on it, we think this deserves a more
> detailed approach, because there are many ways how to integrate the potential
> heuristics there, which, we think, should be discussed and evaluated.

---

### Decision · Program_Chairs · 2021-06-10

**Decision:**

Accept

**Comment:**

Both reviewers strongly recommend to accept the paper. Congratulations on a nicely written paper!